Gender differences in peak medial joint contact forces during activities of daily living

Snyder Samantha J. 1 2
Fakhar Maliheh 1
Shim Jae Kun 1 3 4
Miller Ross H. 1 3 rosshm@umd.edu
1 Department of Kinesiology, University of Maryland , College Park, Maryland , United States
2 Orthopedics and Sports Medicine, MedStar Health Research Institute , Baltimore, Maryland , United States
3 Neuroscience and Cognitive Science Program, University of Maryland , College Park, Maryland , United States
4 Fischell Department of Bioengineering, University of Maryland , College Park, Maryland , United States
Espada Mário
Electronic publication date: 2025 Jul 9
Publication date: 2025
Volume: 13
Electronic Location ID: e19677
Received 2024 Oct 11; Accepted 2025 Jun 9
Copyright: © 2025 Snyder et al.
Copyright year: 2025
Copyright holder: Snyder et al.
License: This is an open access article distributed under the terms of the Creative Commons Attribution License, which permits unrestricted use, distribution, reproduction and adaptation in any medium and for any purpose provided that it is properly attributed. For attribution, the original author(s), title, publication source (PeerJ) and either DOI or URL of the article must be cited.
License URL: https://creativecommons.org/licenses/by/4.0/

Keywords: Gender difference, Osteoarthritis, Activities of daily living, Tibiofemoral

Funding: UM Ventures Medical Device Development Fund at the University of Maryland This work was supported by the UM Ventures Medical Device Development Fund at the University of Maryland. The funders had no role in study design, data collection and analysis, decision to publish, or preparation of the manuscript.

==============================
Women are more likely to suffer from knee osteoarthritis as compared to men. For men and women, greater peak knee medial joint contact force is associated with greater rates of knee osteoarthritis. However, it is unclear if the increased rates of knee osteoarthritis in women is associated with greater medial joint contact force. We hypothesize that because women experience greater rates of knee osteoarthritis, they would experience greater peak medial joint contact force. Fifty-two healthy, young participants (26 women, 26 men) performed sit-to-stand, stand-to-sit, self-selected speed walking, self-selected speed running, and set speed running trials over force plates while motion capture data was recorded. Medial joint contact force, scaled by bodyweight, was calculated with a reduction modeling approach from inverse dynamics data and ultrasound measured distances. Differences in peak medial joint contact force between men and women were tested with one-tailed unpaired Student’s t-tests with a Bonferroni correction. No significant differences were seen between groups peak medial joint contact force in any of the tested movements. Medial joint contact force may not be able to explain the disparity in knee osteoarthritis rates between men and women.

Introduction

Knee osteoarthritis disproportionately affects 4.8% of women globally, compared to 2.8% of men (Cross et al., 2014), and a large proportion of women with knee osteoarthritis experience a lower quality of life, greater pain, and worse knee function compared to men (Dalury et al., 2009; Tonelli et al., 2011). There are biological differences between men and women such as differences in synovial fluid immune cells (Kriegova et al., 2018), hormones (Nevitt et al., 2001; de Klerk et al., 2009) and anatomy (Hame & Alexander, 2013) that may contribute to this disparity (Buckwalter & Lappin, 2000; Kriegova et al., 2018; van Spil & Szilagyi, 2020), but most cases of osteoarthritis can be explained as problems of mechanical overload (Felson, 2013). By comparing mechanical joint loading between men and women, we may stand to gain a greater understanding of factors that can (or cannot) explain gender differences in knee osteoarthritis risk.

Medial tibiofemoral knee osteoarthritis is the most common location for osteoarthritis and is characterized by degradation of articular cartilage on the medial aspect of the knee (Thomas et al., 1975). Mechanically, this loss of cartilage is associated with large repetitive loads on the medial aspect of the knee (Miyazaki et al., 2002; Riemenschneider et al., 2019; Brisson et al., 2021; Amiri et al., 2023). Excessive or unusual loading to the medial aspect of the knee is therefore a suspected catalyst for knee osteoarthritis (Seedhom, 2006; Felson, 2013). Recently, it has been reported that greater peak medial joint contact force (MJCF) relative to bodyweight has been associated with loss of medial tibial cartilage volume (Brisson et al., 2021) and progression of knee osteoarthritis (Amiri et al., 2023). Because women experience greater rates of knee osteoarthritis, it could be expected that women also experience greater peak MJCF compared to men during various activities throughout the day.

Although walking is one of the most common activities during the day, other activities regularly load the knee and are associated with knee pain and knee osteoarthritis (Clynes et al., 2019). Sit-to-stand, stand-to-sit, running, and walking are all activities that occur with a relative high frequency (McLeod et al., 1975; Bohannon, 2015; Bade, Aaron & McPoil, 2016) and relative high peak loading throughout the day (Kutzner et al., 2010; Miller et al., 2014), which would be expected to result in relatively high cumulative damage to the knee cartilage due to the highly nonlinear stress-life relationship in articular cartilage (Riemenschneider et al., 2019). If women take more steps per day than men, this difference could result in greater cumulative knee loads per day as compared to men despite the peak knee loads being similar between the two groups. However, in a 7-day consecutive prospective study including 441 young individuals, there were no significant differences in the total number of steps walked between the two groups (Behrens & Dinger, 2005). Additionally, the current literature does not show any significant differences between the number of sit-to-stand and stand-to-sit activities for women and men (Bohannon, 2015). Therefore, analyzing peak MJCF differences during these activities between young, healthy men and women may give further insight into why women experience greater rates of knee osteoarthritis.

Due to the mechanical progression of knee osteoarthritis, analyzing differences in young, healthy individuals is necessary. Symptomatic knee osteoarthritis prevalence is greater in individuals over the age of 50 (Cross et al., 2014), but the disease does not develop overnight. From a mechanical perspective, the stress-life of cartilage can degenerate due to forces occurring throughout an individual’s lifetime (Weightman, 1976). Greater mechanical loads are associated with greater risk of knee osteoarthritis (Miyazaki et al., 2002; Brisson et al., 2021; Amiri et al., 2023), so younger, healthy individuals that perform daily activities with relatively higher loading may be more susceptible to knee osteoarthritis, particularly beyond the relatively young age of skeletal maturity upon which the collagen matrix ceases significant turnover (Heinemeier et al., 2016). Therefore, although knee osteoarthritis typically becomes symptomatic later in life, younger individuals may already be in a pre-disease stage where subtle changes in cartilage and joint health are beginning to occur.

Kinematic and kinetic gender differences have been examined across various activities (Kerrigan, Todd & Della Croce, 1998; Cho, Park & Kwon, 2004; Bayliss Zajdman et al., 2022), yet there is minimal research on differences in peak MJCF between young, healthy men and women during sit-to-stand, stand-to-sit, walking and running. In recreational runners, men experienced greater MJCF than women during walking and running (Esculier et al., 2017). However, this research did not extend to a non-runner population and did not consider subject- and gender-specific calculations of MJCF. Previously MJCF measurements have relied on fixed average values of tibial plateau widths (Miller et al., 2017) or externally measured knee widths (Esculier et al., 2017). However, using ultrasound with novel analysis techniques to measure femur geometry could provide more accurate, subject and gender-specific measurements of the tibial plateau widths. There are significant differences between male and female individuals muscle moment arms and orientations (Wretenberg et al., 1996), and including these differences could ultimately change knee joint contact force results. The moment arm length is inversely related to the magnitude of a force, so with the same magnitude moment, a women’s smaller moment arm, would result in a relatively larger joint contact force as compared to a man. Additionally, by applying subject specific models, knee contact forces can be more accurately calculated for an individual (Ding et al., 2019).

The present study therefore aims to first establish a novel ultrasound method to estimate subject specific tibial plateau width and secondly determine if young, healthy women experience greater peak MJCF than young, healthy men during self-selected walking, self-selected running, sit-to-stand, and stand-to-sit activities. Because female individuals experience greater knee osteoarthritis rates compared to men, we hypothesized that peak MJCF would be greater in women compared to men.

Materials and Methods

Participants

Fifty-two healthy, young male and female individuals were recruited from the University of Maryland and surrounding community. Participant demographics are shown in Table 1. Participants with a history of major lower limb extremity injuries were excluded. All participants provided written informed consent of protocol, which was approved by the University of Maryland Institutional Review Board (#1335286).

Table 1 Demographic data of participants.

	Female (n = 26)	Male (n = 26)	p	
	Mean ± SD	Mean ± SD		
Age (years)	22.5 ± 3.5	24.3 ± 4.4	0.11	
Weight (kg)	59.9 ± 7.2	71.8 ± 8.7	<0.001	
Height (cm)	162.8 ± 5.9	175.1 ± 6.7	<0.001	
BMI (kg/m2)	22.6 ± 2.2	23.5 ± 2.6	0.21	
Weekly running (miles/week)	2.6 ± 6.0	5.8 ± 14.5	0.36	
Number of recreational runners (miles/week > 0)	14	13		
Note:

Mean and standard deviations (SD) presented.

For 26 participants per group, a power of 0.8, and an alpha of 0.05, with a one-tailed unpaired Student’s t-test the smallest detectable effect size is 0.70. Effect sizes in previous studies relating peak MJCF to knee osteoarthritis progression were 0.75 (Brisson et al., 2021) and 1.22 (Brisson et al., 2021; Amiri et al., 2023). It is unknown what particular effect size of joint loading differences between young men and women are clinically relevant for explaining gender differences in knee osteoarthritis later in life, so it is possible the present study is underpowered for that purpose.

Experimental setup

Participants wore form-fitting clothing and New Balance neutral athletic shoes fitted to the closest whole size (all participants wore the same make and model of shoe). Reflective markers were placed on lower extremities according to locations in Fig. 1. Thirteen Vicon motion capture cameras (Vicon, Oxford, UK) captured marker positions at 200 Hz, and eight six-degree of freedom force plates (Kistler, Winterthur, Switzerland) recorded ground reaction forces at 1,000 Hz. The force platforms were embedded within a raised platform on the lab floor.

Figure 1 Marker placement on a participant.

Example of marker placement locations on a participant.

Ultrasound scanning protocol

Before motion capture data was collected, participants were instructed to lay supine on a table, and the participant’s right knee was positioned at a 140-degree angle (Fig. 2). This angle was measured and confirmed by two researchers with a goniometer, and this standardized angle has been used in previous research imaging knee distal anterior femoral articular cartilage (Battersby et al., 2023). Ultrasound scans were performed with a handheld 12-4 MHz, 34 mm 2D transducer (Lumify L12-4; Philips Healthcare, Orlando, FL, USA) and in accordance to EULAR guidelines (Backhaus et al., 2001). Gel was applied to the ultrasound and skin, and the ultrasound was placed over the medial and lateral condyles of the femur, superior to the patella. The ultrasound probe width is not large enough to image the entire distance, so three videos were taken of the knee. The probe was initially placed on the medial side of the knee and was slid across the knee to the lateral aspect of the knee where the video ended. Between each video, the probe was lifted from the skin and repositioned on the medial aspect of the knee. The ultrasound was in contact with the knee for the entire video without applying excessive pressure. Similar methods using an ultrasound probe to image the distal anterior femur resulted in high intra-rater single image, inter-rater single image, and intra-rater between images reliability (Pamukoff et al., 2018).

Figure 2 Ultrasound placement on a participant.

Depiction of the ultrasound probe placement on the femur superior to the patella. Participant is laying supine with knee at a 140-degree angle.

Study design

Subjects first performed a calibration trial, standing upright with their lower limb joints in neutral positions and their arms in a T-pose. First, participants were directed to perform 10 sit-to-stand and stand-to sit activities with arms crossed on their chest. They were instructed to sit and stand at a pace that matched their usual pace when sitting in a chair during the day. They performed these trials on a stool with a seat height of 44 cm, which falls within the standard chair height regularly used for sit-to-stand tests (Whitney et al., 2005). Next, they performed 10 walking trials at a self-selected pace across the 4.8-m stretch of force plates. Lastly, participants were instructed to run five laps around the 50-m track at a self-selected pace and then five laps at a set speed of 3.0 m/s (Fig. 3). Because men typically run faster than women (Sparlin, O’Donnell & Snow, 1998), a set-speed running condition was included.

Figure 3 Motion capture and force plate data collection layout.

Participant running across force plates while motion capture data recorded. Thirteen motion capture cameras, 4.8-m path, and eight force plates are embedded in a 50-m track.

Data processing

For self-selected walking, self-selected running, and set speed running, an average of five clean strides were extracted from heel strike to heel strike of the right foot. One male participant was excluded from self-selected speed running analysis and one from set speed running analysis due to a minimal number of trials available for those participants. A threshold of 20 N was used in Visual3D software (C-Motion, Germantown, MD, USA) to determine the initiation of heel strike of each walking and running trial. Sit-to-stand and stand-to-sit time phases were based on previously developed definitions and equations (Kralj, Jaeger & Munih, 1990), where both conditions start at the a the onset of a vertical acceleration phase and end at a quiet standing or sitting event. The stand-to-sit period was extracted starting after the quiet standing period and defined by Fz≤0.98 BW, where Fz is the vertical force and BW is the bodyweight. The stand-to-sit phase ends with either the beginning of quiet sitting defined by dFydt≤1%(dFydt)PP, where Fy is the anterior posterior force and PP is the peak-to-peak value or when the pelvis center of gravity velocity in the vertical direction was under 0.02 m/s. The sit-to-stand period was extracted starting at the at the seat unloading phase and defined by dFzdt≥10%(dFzdt)PP. The end of the sit to stand period occurs with the start of quiet standing defined by Fz=(1±0.01) BW.

Markers were labeled in Nexus software (Vicon, Oxfordshire, UK) and were filtered in Visual3D with a 4th order dual pass Butterworth filter with a frequency of 10 Hz for running and 6 Hz for other movements. To filter the ground reaction force data, a 4th order dual pass Butterworth filter was applied with a cutoff frequency of 45 Hz. Previous research analyzing walking and running kinetics applied similar cutoff frequencies (Krupenevich et al., 2015; Hunter et al., 2019; Wang et al., 2023). From the calibration trial marker positions, a linked-segment model of each subject was created. The knee joint center was defined between the medial and lateral femoral epicondyles and ankle joint center between the malleoli markers. These joint centers were later reconstructed as virtual joint centers based on marker positions in the calibration trials. The hip joint center was created using the CODA pelvis in Visual3D with the Anterior Superior Iliac Spine and Posterior Superior Iliac Spine markers (Bell, Brand & Pedersen, 1989). Speed was calculated during each stride in Visual3D using the speed of the center of gravity of the pelvis. Three-dimensional resultant forces and moments were then calculated at the hips, knees, and ankles by inverse dynamics modeling in Visual3D.

Ultrasound image analysis

With OpenCV-Python (version 4.8.1), frames were extracted from each of the three ultrasound videos at ten frames per second, and the resulting frames were stitched together using opencv-python stitcher (Bradski, 2000). The OpenCV-Python stitcher has been used in other ultrasound research to create a single image of segments larger than the ultrasound probe width (Yu et al., 2015). Measurements are taken from each image at the centers of the medial and lateral femoral condyles using a custom Python script where pixel distances were calibrated to centimeters. Two independent researchers measured each image three times (Fig. 4), and the overall mean for each participant was used to directly inform the MJCF calculation. The distance extracted from the ultrasound image was superior to the patella when the knee was flexed at 140 degrees. Since this position of measurement results in an intercondylar distance smaller than the actual midcondylar distance d between the midpoints of the medial and lateral condyles, the measured intercondylar distances were scaled by gender-specific ratios of midcondylar vs. intercondylar distance from cadaver data to determine estimated midcondylar distance d^ for each participant (Terzidis et al., 2012). The mean tibial plateau for female individuals was 4.86 cm and the mean for male individuals was 5.53 cm.

Figure 4 Ultrasound stitched together image.

Example image of stitched together photos from an ultrasound video portraying the medial and lateral condyles of the femur, superior to the patella. The red line with arrows denotes the measured width extracted from the image.

Intraclass correlation coefficients (ICC) were used to assess intra- and inter-rater ultrasound method reliability. Precision was calculated through standard error measurement (SEM). Where, SEM=SD×1−ICC, and SD is the standard deviation. ICC values between 0.75 and 0.9 were considered moderate reliability, while values greater than 0.9 were considered excellent reliability (Koo & Li, 2016). To achieve a minimum ICC of 0.6 and a desired ICC of 0.9, 8 participants were required (Walter, Eliasziw & Donner, 1998). For all tests, the raters were blinded to participant information. The ultrasound examiner established between-day intra-rater reliability and precision by assessing nine participants with the same inclusion/exclusion criteria as the current study separated from two to seven days between measurements. To determine inter-rater reliability between measurements, two raters measured the distances for all 52 participant images three times. To determine intra-rater reliability between scans, one rater measured the distances for all 52 participant images three times and distances across a single image were compared.

Data analysis

Using the inverse dynamics calculated ankle, knee and hip moments and the knee joint resultant forces, the MJCF was calculated using a reduction modeling approach (Morrison, 1968; DeVita & Hortobagyi, 2001) in MATLAB 2021 (The MathWorks, Natick, MA, USA). The reduction modeling approach was based on previously published research and code (Miller & Krupenevich, 2020). This approach calculates individual muscle forces and accounts for muscle co-contraction around the knee, which is important because more severe cases of knee osteoarthritis are associated with increased muscle co-contraction (Heiden, Lloyd & Ackland, 2009). Muscle cross sectional areas were calculated based on data summarized in Miller (2016). First, the gastrocnemius and soleus muscle forces were calculated based on ankle plantar flexion moments. Next, the hamstring and gluteus maximum muscle forces were calculated based on hip extension moments. The previously calculated hamstring and gluteus maximus muscle force contributions were subtracted from the knee flexion moment to calculate the quadriceps force. Moment arms for hip and ankle muscles were expressed as recommended best fitting functions of each joint angles (Menegaldo, De Toledo Fleury & Weber, 2004). These ankle and hip moment arms were then scaled by the average heights of male and female participants to ensure they were gender specific. For example, the male moment arms of the hip were multiplied by the average height of all male participants, 1.75 m, and divided by the average height of all participants, 1.69 m. Moment arms and muscle orientations surrounding the knee were expressed as a quadratic function using average female values for female participants and average male values for male participants (Wretenberg et al., 1996). Lateral collateral, medial collateral, anterior cruciate, and posterior cruciate ligament forces were calculated based on methods in Morrison (1968), and moment arms and orientations for anterior and posterior cruciate ligaments were based on results from previous research (Herzog & Read, 1993). The total MJCF was calculated by summing the moments about the lateral tibiofemoral contact point in the frontal plane (Miller & Krupenevich, 2020):

(1) Fmed=−(∑i=1n⁡riFid^+FRJF2+τKAMd^)

where Fmed is the MJCF, ri and Fi are the moment arm and force of the ith knee muscle or ligament, FRJF is the resultant knee joint force, τKAM is the knee adduction moment, and d^ is the distance between the medial and lateral tibiofemoral contact points. For each subject, the distance d^ was estimated by calculating the distance between the medial and lateral knee condyles using ultrasound images of the knee. The resulting MJCF value was scaled by bodyweight (BW) to account for significant gender-based differences in bodyweight between the two groups. The dynamic alignment and the long axis of the tibia were included in the modeling via the motion capture data and subject models in Visual3D.

Because knee adduction moment (KAM) and knee flexion moment (KFM) are key inputs to the MJCF calculation and have also been linked to knee osteoarthritis development and progression (Miyazaki et al., 2002; Chehab et al., 2014), we also extracted peak KAM and KFM for comparison between groups. For analysis, these moments were scaled by height (Ht) and by BW (Miyazaki et al., 2002; Chehab et al., 2014).

Statistical analysis

For each participant, peak KAM, peak KFM, peak MJCF, and speed, were averaged over all trials for each condition. Normality was tested using a Shapiro-Wilks test and variance between datasets was tested with Levene’s test. If data did not come from a normal distribution or there was significant variance between male and female data, a Mann-Whitney-U test was applied to test significance between groups. Otherwise, a one-tailed unpaired Student’s t-test was applied to test if there was a difference between groups. As men typically run faster than women (Sparlin, O’Donnell & Snow, 1998), a one-tailed unpaired Student’s t-test was used to test differences in self-selected running speeds (α = 0.05). Meanwhile, a two-tailed unpaired Student’s t-test was used to test differences in self-selected walking speeds (α = 0.05). These tests were conducted using SciPy library (version 1.9.3) (Virtanen et al., 2020). Effect sizes, mean change divided by pooled standard deviation, was calculated as Cohen’s d. After a Bonferroni correction, significance for differences in peak knee loads statistical tests was set to α = 0.0028. The Bonferroni correction was set by dividing the alpha of 0.05 by the 18 statistical tests. The 18 tests include five conditions (walk, run, set speed run, sit-to-stand, stand-to-sit) and three variables (MJCF, KAM, KFM) as well as the three tests for speed (walk, run, set speed run). This results in a Bonferroni correction of 0.0028.

Results

Reliability of ultrasound measures are presented in Table 2. Reliability of ultrasound methods ranged from moderate (ICC(2,3) = 0.93) to excellent (ICC(2,3) = 0.99) per Koo & Li (2016) definitions.

Table 2 Reliability analysis of ultrasound methods.

	ICC (2,3)	95% CI	SEM (mm)	
Intra-rater: one rater, across two sessions	0.93	[0.71–0.98]	1.03	
Inter-rater: two raters, three images	0.99	[0.98–0.99]	0.44	
Intra-rater: one rater, three images	0.95	[0.92–0.97]	0.97	
Note:

Intraclass correlation coefficients (ICC), 95% confidence intervals (CI), and standard error of measurement (SEM) presented in table for each type of reliability tested.

Average values for peak MJCF, KFM, and KAM for the various tested movements in men and women are presented in Fig. 5. Percent differences in peak values are listed for each activity in Table 3. Walking speed did not differ between men and women (men: 1.31 ± 0.19 m/s, women: 1.28 ± 0.18, p = 0.75, Cohen’s d = 0.20). Although men ran 9% faster than women, the difference in self-selected speed was not statistically significant (men: 2.97 ± 0.56 m/s, women: 2.71 ± 0.37, p = 0.041, Cohen’s d = 0.74). There were no differences between men and women in peak MJCF, peak KFM, and peak KAM for any of the movements (all p ≥ 0.11).

Figure 5 Peak knee load differences in men and women.

Peak medial joint contact force (MJCF), peak knee flexion moment (KFM), and peak knee adduction moment (KAM) for men and women during self-selected walking, self-selected pace running, 3 m/s running, stand-to-sit, and sit-to-stand activities.

Table 3 Percent differences between men and women peak knee loads.

	MJCF (%)	KAM (%)	KFM (%)	
Walk peak 1	7.7	8.9	3.9	
Walk peak 2	0.7	9.2	15.8	
Run	11.5	22.9	3.8	
3 m/s Run	8.6	24.3	0.7	
Stand-to-sit	11.7	6.0	4.6	
Sit-to-stand	12.5	7.7	6.7	
Note:

Percent differences between men and women peak medial joint contact force (MJCF), knee adduction moment (KAM), and knee flexion moment (KFM) during each activity.

Discussion

The present study developed a novel ultrasound protocol with moderate to excellent reliability and examined differences in peak MJCF between young, healthy men and women during self-selected walking, self-selected running, set speed running, sit-to-stand, and stand-to-sit activities. Our hypothesis, that women would experience greater peak MJCF than men, was not supported. Since knee osteoarthritis is thought to develop and progress at least in part from placing high peak loads on the knee in movements like walking (Miyazaki et al., 2002; Amin et al., 2004; Lynn, Reid & Costigan, 2007; Chehab et al., 2014), these results suggest that peak knee loads when young do not explain why women have a greater prevalence of knee osteoarthritis later in life.

There were no significant differences between men and women in peak MJCF during walking. In related research, Esculier et al. (2017) found +0.1 BW greater peak MJCF for recreational runner men compared to women when both groups walked at a set speed of 1.3 m/s on a treadmill. The difference in walking peak MJCF findings, with no significant difference here even though the average peak was +0.2 BW greater in women, could partially be attributed to the set speed in treadmill walking vs. the self-selected speed in overground walking, leading to more variance in the present data. The sample size in Esculier et al. (2017) was also considerably larger than the present study (87 vs. 52), allowing for smaller effects to be detected as significant. Neither of these differences, ~0.1 BW greater in men or ~0.2 BW greater in women, are likely large enough per se to present a substantive risk for knee OA.

By a similar rationale, the present results on running are roughly comparable to the report from Esculier et al. (2017), with men running at slightly faster self-selected speeds and MJCF peaks (+0.8 BW in men vs. women) that did not reach statistical significance in this smaller sample with more variable overground locomotion (the difference of +0.6 BW in men was significant in the previous study).

Additionally, the MJCF model applied in Esculier and colleagues research did not take into account significant differences between men and women muscle moment arms and muscle orientation (Wretenberg et al., 1996). We used gender specific muscle insertion angles, knee muscle moment arms, and subject and gender specific tibial plateau widths. Gender specific muscle parameters can significantly alter results in knee joint contact forces as compared to a general model (Ding et al., 2019). Our gender- and subject-specific considerations may also be why our results differed from previous research during walking (Esculier et al., 2017).

We found no differences in sit-to-stand and stand-to-sit peak MJCF between male and female individuals. We used the same seat height for both groups. This seat height reflected the typical chair height used for sit-to-stand tests (Whitney et al., 2005), and throughout a day, there are many instances where sitting and standing from a set height chair are required. Taller individuals typically need a deeper squat to sit at the same height as shorter individuals, which could affect knee loads (Nagura et al., 2002). Although men were taller on average, we found no differences in MJCF between groups for this activity. No previous research analyzed differences in peak sit-to-stand and stand-to-sit MJCF between men and women, and future research could analyze the effect of seat height on this variable between groups.

Despite the gender- and subject-specific elements of the present knee model, there are of course some notable limitations to the present work. For example, unlike the knee moment arms, the hip and ankle muscle moment arms reported in literature were not gender-specific, which affects the knee MJCF results through the biarticular muscles and the model’s co-contraction assumptions. This effect is likely rather small given the typical relative timing in the gait cycle of hip flexor, quadriceps, and ankle plantar flexor muscle activity (Novacheck, 1998; Sutherland, 2001), but based on the timing of muscle activations, the gastrocnemius muscle force would affect the calculations of MJCF for the second peak during walking (Sutherland, 2001). In order to account for this, we scaled the ankle and hip muscle moment arms by relative heights of male and female participants. With the ultrasound protocol, we used measured femur geometry to inform the key model parameter of the distance between the femoral condyle contact points on the tibial plateau. The condyle widths were then scaled subject-specifically from the generic tibial plateau cadaver measurements. This scaling method, with set knee widths rather than ultrasound widths, has been implemented previously to individualize models by scaling external knee widths to the average knee tibial plateau cadaver values (Miller et al., 2017). Future research can improve our ultrasound informed model by establishing the relationship between the distal anterior femoral head and the actual bicondylar width with MRI measurements.

Due to the relationship between KFM, KAM, and MJCF, as well as the importance of KAM and KFM in knee osteoarthritis progression and development, we did an exploratory analysis of peak KFM and KAM differences between groups. We found no differences in peak KAM and KFM across all activities between male and female individuals. The similarity in peak KAM between men and women was expected based on previous research. We found no significant differences in running peak KAM between men and women, which agrees with previous research examining recreational, young runners running at self-selected speeds (Esculier et al., 2017). Our walking results are consistent with previous studies looking at adults in the United States walking barefoot and healthy Korean adults walking at self-selected speeds (Kerrigan et al., 2000; Cho, Park & Kwon, 2004). Although some studies found that women walked with significantly greater peak KAM (Esculier et al., 2017; Obrębska, Skubich & Piszczatowski, 2020), walking was not analyzed at self-selected speeds in these studies. Obreska and colleagues instructed participants to walk at self-selected speeds but analyzed only trials where a specific speed was reached. Esculier and colleagues’ participants were recreational runners who walked at a set speed of 1.3 m/s on a treadmill, and as mentioned earlier, this matches our self-selected speeds, but our participants walked overground. Our research was the first to analyze young, healthy individuals performing stand-to-sit and sit-to-stand activities in a standard height chair and we found no differences between group peak KAM. Research of older age groups found differences between men and women in peak KAM during squatting (Bayliss Zajdman et al., 2022) and chair rise activities (Luepongsak et al., 2002), so future research should consider age when examining knee loading gender differences during various activities.

We also found no difference between peak KFM between the groups. Esculier and colleagues found at self-selected running speeds, male recreational runners ran with greater peak KFM than the female runners (Esculier et al., 2017), while a study looking at set speed running of recreational runners, 3.65 m/s, there were no differences in peak KFM between groups (Ferber, Davis & Williams, 2003). For walking, one research study reflects our peak KFM results, where no differences were found between healthy, young men and women walking at self-selected speeds barefoot (Kerrigan et al., 2000). However, another study looking at young, healthy Korean female individuals walking at self-selected speeds had greater second peak KFM (Cho, Park & Kwon, 2004). Cho, Park & Kwon (2004) did find that Korean adults walked with some differences in kinematics and kinetics as compared to US based individuals, which could be why they found differences in KFM between groups. Meanwhile, other research not at self-selected speeds has varying results where some studies saw women had a greater first peak KFM (Obrębska, Skubich & Piszczatowski, 2020) or men experiencing a greater first peak KFM (Esculier et al., 2017). Our sit-to-stand and stand-to-sit activities reflect previous results where there were no differences shown in KFM in older adults during sit-to-stand and stand-to-sit and no differences in KFM during squatting (Luepongsak et al., 2002; Bayliss Zajdman et al., 2022).

This research examined the difference in mechanical knee loads because of the association between greater knee loads and greater risk and progression of knee osteoarthritis (Miyazaki et al., 2002). Focusing solely on mechanical loads is relevant because if differences in mechanical loads were to exist, interventions to reduce these loads could be implemented to improve knee function and reduce osteoarthritis risk (Shull et al., 2013). There are many non-mechanical factors that may contribute to the gender-based discrepancies in knee osteoarthritis rates, such as differences in immune cells in synovial fluid between genders (Kriegova et al., 2018). Furthermore, hormonal differences, such as effects of estrogen on cartilage metabolism and joint health (Nevitt et al., 2001; de Klerk et al., 2009) and anatomical differences, including variations in joint alignment (Hame & Alexander, 2013), may also contribute to the higher incidence of knee osteoarthritis in women. Including these factors in future research, alongside mechanical loads, may provide a more comprehensive understanding of the differences in knee osteoarthritis rates between men and women. Additionally, knee osteoarthritis occurrence increases dramatically after age 45 (Deshpande et al., 2016), and obese individuals (BMI > 30) are at higher risk of the disease (Kulkarni et al., 2016). The average age of our participants was 23 years, and the average BMI was 23 kg/m2. A more comprehensive analysis that accounts for factors like age, obesity, and gender could offer a clearer understanding of whether differences in peak knee loads between males and females become apparent in middle age or at higher body weights. Additionally, a future longitudinal study exploring the interplay between gender, knee joint loading, knee osteoarthritis, and other biological factors could help clarify whether knee loading in combination with additional risk factors contributes to the higher incidence of the disease observed in women.

Although there were no differences in peak MJCF, KAM, and KFM, the male participants were significantly taller and heavier than the female participants. Scaling for these size-related anatomical factors renders data like joint moments in a form where differences can be more confidently related to the mechanics of the movement vs. differences in size, and thus biomechanical risk factors of knee osteoarthritis are typically scaled. KAM and KFM are generally scaled by bodyweight and height, while MJCF is typically scaled just by bodyweight. Taller individuals will have longer moment arms, leading to proportionally larger moments. Larger individuals will typically experience greater absolute forces on the joints, but presuming articular cartilage can be conditioned to larger loads (Arokoski et al., 2000; Seedhom, 2006), scaling by bodyweight accounts for presumed natural adaptations in the cartilage to larger loads, and the great majority of studies that have related relatively large magnitudes of knee loads and moments to osteoarthritis have used magnitudes scaled by bodyweight, and in many cases for moments, also by height. Overall, scaling by height and weight helps mitigate the effects of differences in size across participants (Moisio et al., 2003), and the results of data like joint moments and loads can then be more confidently attributed to differences in movement mechanics between the two genders vs. differences in body size that may or may not be purely gender-related (e.g., larger men would have greater absolute MJCF than smaller men).

Conclusions

Overall, our results suggest that peak KAM, peak KFM, and peak MJCF when young do not explain the disproportionate number of women affected by knee osteoarthritis. There are other non-mechanical factors that were not considered in this research and could affect the development of knee osteoarthritis such as cellular, hormonal, and anatomical factors (Nevitt et al., 2001; de Klerk et al., 2009; Hame & Alexander, 2013). A comprehensive analysis of the biomechanical and non-biomechanical factors in healthy young male and female individuals could improve the understanding of disease progression in both groups. Additionally, knee osteoarthritis occurrence increases in older adults (Deshpande et al., 2016) and obese individuals (Kulkarni et al., 2016), while our participants were on average relatively young with healthy BMIs. An analysis with confounding factors such as age and obesity with gender may better inform if there are any peak KAM or MJCF differences between male and female individuals, e.g., if differences emerge in middle-age or at heavier bodyweights.

Additional Information and Declarations

Competing Interests

Ross H. Miller is an Academic Editor for PeerJ.

Author Contributions

Samantha J. Snyder conceived and designed the experiments, performed the experiments, analyzed the data, prepared figures and/or tables, authored or reviewed drafts of the article, and approved the final draft.

Maliheh Fakhar performed the experiments, authored or reviewed drafts of the article, and approved the final draft.

Jae Kun Shim conceived and designed the experiments, authored or reviewed drafts of the article, supervision, and approved the final draft.

Ross H. Miller conceived and designed the experiments, authored or reviewed drafts of the article, supervision, and approved the final draft.

Human Ethics

The following information was supplied relating to ethical approvals (i.e., approving body and any reference numbers):

Protocol was approved by the University of Maryland Institutional Review Board (Ethical Application Ref: #1335286).

Data Availability

The following information was supplied regarding data availability:

The raw data is available at figshare: Snyder, Samantha (2024). Data to accompany “Gender Differences in Peak Medial Joint Contact Forces during Activities of Daily Living”. figshare. Dataset. https://doi.org/10.6084/m9.figshare.27196251.v2.

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
