# Peer review of "Gender differences in peak medial joint contact forces during activities of daily living"

_PeerJ, doi:10.7717/peerj.19677_

## Round 0.1 · original submission · Major Revisions

· Academic Editor

Major Revisions

Dear Authors,

Please revise the manuscript considering the reviewers´ suggestions.

Thank you.

Best regards.

Reviewer 1 ·

Basic reporting

This manuscript deals with the medial joint contact forces in men and women during ADL. The manuscript it written fine however, rather long, now and then some redundancy appears.

80 references are a lot for a normal research paper.

The authors integrate a reliability analysis in the manuscript, however the results are presented in the methods. Also, the results of the US measurements are presented in the methods, although this is part of the experimental study and results could be expected. So please restructure your manuscript or publish the US results in another manuscript.

Experimental design

I have some major problems with the current setup. First of all, did the authors not take the leg axis into account. It is known that women have a more valgus leg axis. Valgus malalignment can reduce the medial joint contact force, which might explain why the authors did not find any differences.

Further, I doubt if young healthy subjects, 25 years of age, are the best group to study OA. You are only measuring one possible parameter. In time between onset (20 years) a lot can happen and as you mention there are a lot of influences. The only way to know if these persons become OA is to follow them for the next 40 years.

The methods appear to be used before, however it reads if it is all put together for the first time. It does not become clear how the MJCF are calculated and how the ultrasound measurements contribute to the calculation.

Validity of the findings

Statistics see my comments under 4

Conclusion contains information which should have been mentioned in the introduction

Additional comments

Introduction
Line 53: please explain which biological and physical differences might contribute.
Line 100: please explain how muscle orientation differ between men and women and how this can change the muscle force.

Methods
Line 187: please add a reference to the CODA pelvis protocol.
Line 202-215: this should be part of the result section. However, it could also be the question if the whole US reliability should be part of this paper….?
Line 196-215: It does not become clear what the parameters of interest are in the US protocol.
Line 220: did you mean knee joint resultant forces?
Line 223-226. Please rephrase this sentences.
Line 244-250: It does not become clear how the US measurements are integrated in the calculations. Is the cadaver from Terzidis in the model the authors are using? And why are mean values for men and women calculated? Why not use the individual values?
Line 250-252: The step how the MJCF are calculated is missing.
Line 265: why is here a one-tailed test chosen?
Line 272-173: please explain the Bonferroni correction. Which and how many tests?

Results
When all the measured parameters are mentioned in the methods, the results for the US measurement should be added here as well.
Line 278-280: add m/s as unit.

Discussion
The discussion is rather long.
Line 333-228: this is the rationale for your US methods so why does it appear only in the discussion? Should have been in the introduction why it is important and in your methods how it has been applied.
Line 314: please add a reference for this relationship.

Conclusions
Line 399-403: this should have been part of your introduction.

Reviewer 2 ·

Basic reporting

This study compares knee joint contact forces during activities of daily living in healthy young men and women. Background reviews previous studies focusing on knee joint contact forces as a contributing factor to knee osteoarthritis. However, the subjects in this study were only healthy young adults and had no knee joint disorders. Esculier JF et al. have similarly examined mechanical stress of the knee joint in healthy young subjects, but have paid less attention to disease or disability in the background. It is necessary to demonstrate the rationale for targeting healthy young adults, and if osteoarthritis of the knee is to be assumed, it is preferable to target middle-aged and older adults or others who have some physiological change in the knee joint.

Experimental design

Regarding the number of subjects, a power analysis was used, but the literature referred to is based on patients with osteoarthritis of the knee, and it is unclear whether it can be used in the same way in this study, which is based on healthy young subjects. As a result, there were fewer subjects than in Esculier JF et al. and no statistically significant differences between groups. The hypothesis could be tested by modifying the experimental design.

Validity of the findings

The results are clearly stated and considered reliable. On the other hand, the power of the study is as just described, and it is unclear whether the study design is sufficient to test the hypotheses the authors attempt to make.

Additional comments

If, as the authors claim, mechanical factors alone cannot explain sex differences in knee osteoarthritis, then physiological factors such as hormones should also be mentioned in the discussion.

---

## Round 0.2 · Major Revisions

· Academic Editor

Major Revisions

Dear Authors,

Please further revise the manuscript considering the Reviewer 1 indications.

Thank you.

Best regards.

Reviewer 1 ·

Basic reporting

The authors responded that they removed redundant references; however, there are still 73 references. For a normal research paper, this is still a lot but I will leave it up to the editor.

Introduction: please shorten and structure the introducion

Discussion: please structure and shorten the discussion

Figures: please separate and title and additional information which should be wirtten under the figure

Experimental design

I think the ultrasound protocol should be written as an extra aim in the introduciton. It will make it easier to structure the manuscript.

Validity of the findings

I stil l have my doubt whether with the young individuals you can conclude that mechanical loads does not explain the higher prevalence in women at older age.

Additional comments

Comment to responses
Basic reporting
The authors responded that they removed redundant references; however, there are still 73 references. For a normal research paper, this is still a lot but I will leave it up to the editor.

Experimental design
The authors answer that leg alignment is incorporated in the modeling. Please add a sentence to the modeling approach that personal models were created which took the individual leg alignment into account.
If so, you do not need to add line 323-325 in the statistical analysis section. It was just not clear that leg alignment was taken into account in the model.

The authors have added some information on why it is important to study young individuals. However, they mention that younger individuals might already be in a pre-disease stage with changes in cartilage and joint health. Then why are such parameters not measured? In addition, a long-time follow-up study is not discussed or mentioned as a study setup.

Additional comments
The authors write that the ultrasound technique is novel, should this not be part of the research question to establish a valid protocol to measure the knee width?

Comments on manuscript
Table 1: Why not write p<0.001 instead of 1.9e-6?
Mean and standard deviations (SD) are presented should be written below the table as it is not part of the title.

Table 2: Move the information, which is not part of the title under the table.

Figure 5: if the values are differences, the y axis should be delta peak MJCF. The values are absolute values the title should be changed to peak knee load in men and women.

Figure 1 / Materials and methods
Why did you chose to put the markers on the clothing and not directly on the skin?
You were in a lab right?
What marker movement errors did you experience with this setup?

Line 29: rates are

Line 30-32: This is a general statement and cannot be applied to your study group because they do not have OA

Line 39-40: Maybe they do, but not alone and not in your study population.

Line 73-105 this section is very long, please shorten.

Line 83-85 and line 92-93. In line 80-83 the authors highlight the importance of the frequency of performance. In the following line only the magnitude is mentioned, whereas without knowing the frequency (is there are difference between men and women?), it is difficult to relate load in women to an increase in OA.

Line 106: here the motivation for your US measurements is explained. You should add the setup of the protocol to the aims of your study.

Line 141-144: is this not the aim of your study? Here you state the study is likely underpowered for this purpose. Should this not be mentioned in the limitations of your study, instead of in the methods section?

Line 164-165: what do you mean by the probe was removed from the skin and replaced on the medial aspect of the knee? How does this work when you want the knee measured in 3 videos?

Line 172-173: you can delete this sentence since it is written in line 175 to 182 as well.

Line 174-175: move this sentence to line 180

Line 226-227: combine with line 225-226 …was used for data analysis and to inform the MJCF calculation.

Line 259: quadriceps

Line 261-265: Individual knee width values are calculated. Moment arms are scaled to gender. Why not scale the moment arms to the individual subject?

Line 270-272: summing the moments about the lateral tibiofemoral contact point in the frontal plane. The moments in frontal plane? The contact point?
Why not include the moments in the sagittal plane as well? There is literature that best predict contact forces based on a combination from frontal and sagittal moments.

Line 289-294: since this is part of the method to calculate knee width I would move this section to ultrasound image analysis (line 227). Because here it says the the overall mean will be used to directly inform the MJCF calculations. This is not true when an additional scaling is applied.

Line 297: where is KFM Eq 1?

Line 301: why scaled by height and weight? Please add a reference.

Line 304: delete outcomes variables

Line 322-323: delete peak KAM, peak KFM, and peak MJCF. Since you have more variables like speed, you cannot just mention 3 of them.

Line 323-325: if it is clear that you take leg alignment into account in the model, you do not need this sentence.

Line 331: explain the correction why you come to x = 0.0028 here as well, like you did in the response to the reviewers.

Line 334-335: please add the mean values of the knee width here (you deleted them from line 289). Add the range of reliability in the text as well.

Line 339: speed

Line 341: …. between men and women in peak MJCF, peak KFM and peak KAM for ….
Line 336-342: Please add some information on how high the differences are. From figure 5 it is hard to get values. Express in % or absolute values.

Discussion: please structure and shorten the discussion

Line 410: was expected

Line 446-464: this all comes back in the conclusion.

Reviewer 2 ·

Basic reporting

No problem.

Experimental design

No problem.

Validity of the findings

No problem.

Additional comments

No problem.

---

## Round 0.3 · accepted · Accept

· Academic Editor

Accept

Dear Authors,

I would like to extend my heartfelt thanks for your valuable contribution during the review process and to congratulate you all on the work you've done on this article.

Best wishes for continued excellent work in the future.

Thank you.

Best regards.

Reviewer 1 ·

Basic reporting

ok

Experimental design

ok

Validity of the findings

ok

Additional comments

ok